# Zebrafish Melanoma-Derived Interstitial EVs Are Carriers of ncRNAs That Induce Inflammation

**DOI:** 10.3390/ijms23105510

**Published:** 2022-05-14

**Authors:** Valentina Biagini, Federica Busi, Viviana Anelli, Emanuela Kerschbamer, Marta Baghini, Elena Gurrieri, Michela Notarangelo, Isabella Pesce, Guillaume van Niel, Vito G. D’Agostino, Marina Mione

**Affiliations:** 1Experimental Cancer Biology, Department of Cellular, Computational and Integrative Biology (Cibio), University of Trento, 38123 Trento, Italy; valen165@gmail.com (V.B.); federica.busi@unitn.it (F.B.); viviana.anelli@gmail.com (V.A.); baghinim@yahoo.it (M.B.); 2Bioinformatic Facility, Department of Cellular, Computational and Integrative Biology (Cibio), University of Trento, 38123 Trento, Italy; emanuelaker@gmail.com; 3Laboratory of Biotechnology and Nanomedicine, Department of Cellular, Computational and Integrative Biology (Cibio), University of Trento, 38123 Trento, Italy; elena.gurrieri@unitn.it (E.G.); michela.notarangelo@unitn.it (M.N.); vito.dagostino@unitn.it (V.G.D.); 4Cell Analysis and Separation Core Facility, Department of Cellular, Computational and Integrative Biology (Cibio), University of Trento, 38123 Trento, Italy; isabella.pesce@unitn.it; 5Institute of Psychiatry and Neuroscience of Paris (IPNP), INSERM U1266, Université de Paris, F-75014 Paris, France; guillaume.van-niel@inserm.fr; 6GHU Paris Psychiatrie et Neurosciences, Hôpital Sainte Anne, F-75014 Paris, France

**Keywords:** long ncRNA, P and MRP RNAse, melanoma, *danio rerio*, exosomes/extracellular vesicles

## Abstract

Extracellular vesicles (EVs) are membranous particles released by all cell types. Their role as functional carrier of bioactive molecules is boosted by cells that actively secrete them in biological fluids or in the intercellular space (interstitial EVs, iEVs). Here we have optimised a method for the isolation and characterization of zebrafish iEVs from whole melanoma tissues. Zebrafish melanoma iEVs are around 140 nm in diameter, as determined by nanoparticle tracking and transmission electron microscopy (TEM) analysis. Western blot analysis shows enrichment for CD63 and Alix in the iEV fraction, but not in melanoma cell lysates. Super resolution and confocal microscopy reveal that purified zebrafish iEVs are green fluorescent protein positive (GFP+), indicating that they integrate the oncogene GFP-HRAS^V12G^ used to induce melanoma in this model within their vesicular membrane or luminal content. Analysis of RNA-Seq data found 118 non-coding (nc)RNAs differentially distributed between zebrafish melanoma and their iEVs, with only 17 of them being selectively enriched in iEVs. Among these, the RNA components of RNAses P and MRP, which process ribosomal RNA precursors, mitochondrial RNAs, and some mRNAs, were enriched in zebrafish and human melanoma EVs, but not in iEVs extracted from brain tumours. We found that melanoma iEVs induce an inflammatory response when injected in larvae, with increased expression of interferon responsive genes, and this effect is reproduced by MRP- or P-RNAs injected into circulation. This suggests that zebrafish melanoma iEVs are a source of MRP- and P-RNAs that can trigger inflammation in cells of the innate immune system.

## 1. Introduction

Over the past years, extracellular vesicles (EVs) have emerged as key players in paracrine cell-to-cell and cross-organ communication for their ability to function as carriers of biomolecules (e.g., lipids, proteins, RNA, and DNA) that can be horizontally transferred to recipient cells [1,2,3]. Indeed, they have been recognised to participate in crucial physiological and pathological processes, including viral infections, CNS-related diseases, and cancer, with either disease-promoting or restraining functions that are highly context-dependent. Recent technical advances, especially related to the development of high-resolution imaging techniques (e.g., confocal microscopy, correlated light and electron microscopy-CLEM, super-resolution microscopy) and the refinement of multiple EV isolation methods (e.g., ultracentrifugation, ultrafiltration, size-exclusion chromatography, immunoaffinity-based methods, and nickel-based isolation—NBI) have pushed towards a more appropriate characterization of these heterogeneous nanoscale structures [4,5,6,7].

In fact, both EV size and biological content appear highly variable and may differ according to the cell of origin, state, and environmental stimuli [3]. Generally, EVs can be subdivided into three major categories: ectosomes, exosomes, and apoptotic bodies. Ectosomes (i.e., microvesicles, microparticles) originate from direct budding of the plasma membrane and retain a wide size range that spans from ~100 nm to 1 μm in diameter. Exosomes, whose biogenesis has been characterized more deeply and implies the fusion of multivesicular bodies (MVB) containing intraluminal vesicles (ILVs) with the plasma membrane, have a narrower size range, which is between 40 and 160 nm in diameter. Apoptotic bodies, large vesicles with diameters between 1000 to 5000 nm, are released by apoptotic cells during the late stages of apoptosis [8,9,10].

Numerous studies have highlighted how EVs can be differentially involved in several classical cancer hallmarks, positively influencing tumour growth, metastasis, and therapeutic resistance [1,11]. These aspects are notably important considering several reports demonstrating the massive release of these vesicular structures by cancer cells [12,13,14]. Indeed, it has been demonstrated that neoplastic reprogramming can be driven by the EV-mediated transfer of microRNAs from neoplastic to other cells [15,16]. Still, a well-recognised contribution of EVs to cancer is their ability to influence distinct aspects of the metastatic process. This is achieved by directly influencing the phenotype of cancer cells, by enhancing their metastatic potential, and/or through the education of stromal cells present within the tumour microenvironment (TME) promoting the establishment of a premetastatic niche, which ultimately favours metastatic cell growth in secondary organs [17,18,19,20].

Related to this, most of the attention in characterizing EV function and content has been restricted to circulating EVs. This is due to the fact that these membranous structures are released in all biological fluids, an aspect that opens up the possibility to easily perform liquid biopsies and exploit them as potential biomarkers for diagnostic purposes in different pathological conditions [1,21]. However, EVs play crucial roles also in paracrine cell-to-cell communication and are present in the interstitium of different organs and tissues [22,23,24]. Still, demonstrations of such functions have been hampered by the limited availability of methods for the isolation of interstitial EVs (iEVs) from whole tissues [25,26].

In the present work, we took advantage of a zebrafish model of melanoma [27] to isolate and characterize zebrafish iEVs from whole melanoma tissues without altering the integrity of dissociated melanoma cells. Exploiting the recently-described NBI method [7] as an approach that could preserve the integrity and stability of a polydisperse EV-containing suspension, we integrated some modifications tailored for the zebrafish model used in the study. Following this optimization, we were then able to effectively characterize zebrafish melanoma iEVs for their concentration, size, morphology, and EV-related markers. Additionally, through confocal and super-resolution microscopy, we showed that purified iEVs are GFP+, which results from cargo of the GFP-tagged RAS oncogene present in the tumour tissue. Notably, this observation allowed for easy tracking of zebrafish melanoma iEVs without pre-labelling or post-isolation steps with fluorescent dyes. To further characterize the recovered membranous particles, we performed Next Generation Sequencing (NGS) to identify the populations of ncRNAs present in zebrafish melanoma iEVs as compared with the tissue of origin. This analysis revealed a specific accumulation of the RNA components of P and MRP RNAses in melanoma iEVs. To functionally characterize them, we used an autologous-injection model and showed that melanoma iEVs induce immune responses mediated by the effects of their ncRNA content on tissue macrophages.

## 2. Results

### 2.1. Interstitial Extracellular Vesicles (iEVs) Can Be Isolated from Zebrafish Melanoma

To investigate the release of EVs within zebrafish melanoma tissue, zebrafish melanomas were first dissected from kita:RAS fishes; tissue was dissociated to single cells using collagenase and then submitted to EV capture following the NBI procedure [7] (Figure 1A). As these EVs were extracted from solid cancer tissue with a gentle digestion that preserved the integrity of melanoma cells (Appendix A), they were named interstitial (i)EVs. Of note, purified zebrafish iEVs were found to be GFP+, indicating that these structures likely integrate GFP-RAS within their vesicular membranes or in their lumen, since in this transgenic model, the oncogene is fused to the GFP protein. Zebrafish iEVs were then visualized through confocal microscopy and super resolution microscopy (Figure 1B, Appendix A). Super-resolution images were acquired in TIRF mode, which allowed for the excitation of fluorophores in an extremely thin axial region, thus reducing background noise. Analysis with imaging flow cytometry (Figure 1C) revealed that the majority of iEVs corresponded to small round particles fairly homogenous in size (less than 200 nm) and shape, with at least 45% marked by GFP expression (Figure 1D, Appendix A).

Purified iEVs were then characterized for their size and concentration using the Nanosight instrument. Results indicated that the concentration of iEVs retrieved from kita:RAS melanomas ranged between 3 × 10^8^ and 7 × 10^8^ particles/mL (corresponding to 7 × 10^10^ and 1.6 × 10^11^ particles/10^6^ cells), suggesting a massive release of iEVs from zebrafish tumour tissue. For comparison, we obtained about three- to six-fold fewer EVs starting from the same number of A375M melanoma cells, i.e., 1.3–1.9 × 10^8^ particles per mL (corresponding to 1.2–1.7 × 10^10^ particles/10^6^ cells, Figure 1 E–G). Nanoparticle tracking analysis (NTA) showed that size distribution was heterogenous within the recovered iEV populations. The mean diameter of recovered iEVs was in the range of 200–220 nm, with a mode of about 140 nm, indicating the prevalence of small EVs in the isolated samples (Figure 1E,G). We also characterized iEVs using qNANO, and with this approach we found that the number of small iEVs was in a similar range as that obtained with NTA, i.e., from 1 × 10^8^ to 8 × 10^8^ particles/mL. On the other hand, the number of large iEVs in different samples was more homogeneous, in the range of 2–4 × 10^8^ particles/mL (Appendix A).

Overall, these results indicate that zebrafish melanomas produce and release a large quantity of iEVs, mostly in the size range of exosomes.

### 2.2. Characterization of Zebrafish Melanoma iEVs

To better characterize iEV preparations from the interstitial tissue of zebrafish melanomas, we performed transmission electron microscopy (TEM).

As illustrated in Figure 2A, which shows a representative electron microscopy (EM) section, extracellular vesicles could be identified in the intercellular space and appeared to be entrapped between cells. We also used TEM to visualize zebrafish melanoma iEVs isolated either with the NBI method or with ultracentrifugation. Analysis of electron micrographs obtained with both methods revealed the presence of heterogenous particles resembling EVs both in size and shape (Figure 2B,C). Isolation with the NBI method allowed us to obtain cleaner iEV preparations compared to the ultracentrifugation method. 

To further confirm the identity of isolated particles, western blot (WB) analysis was performed according to the *Minimal Information for Studies of EVs* (MISEV) report [28]. WB analysis of 1 ug total proteins of each fraction demonstrated the enrichment of the exosome-related markers Alix and TSG101 in the iEV fraction, which were barely detectable in the whole melanoma cell lysate. In contrast, the ER marker, calnexin, was enriched in the melanoma lysate and absent in the iEV fraction, thus confirming the presence of exosomes in the polydisperse zebrafish iEVs population isolated with the NBI method (Figure 2D). As a marker for both fractions we used GFP, which marked the GFP-HRAS fusion protein and showed it to be present in both.

Thus, transmission EM and WB analysis for exosome markers confirmed that zebrafish melanoma iEVs present morphological and biochemical features of extracellular vesicles, including exosomes.

### 2.3. Analysis of ncRNA in iEVs and Melanoma Reveals Enrichment of 17 ncRNA Species in iEVs

To better understand the molecular features of melanoma iEVs, we characterized the non-coding RNAs (ncRNAs) present in zebrafish iEVs and melanoma tissue by RNA-Seq analysis. Starting from 1 mg of tumour tissue, we extracted total RNA from purified iEVs and from tissue checked the quality of RNAs at the bioanalyser (Appendix A) and prepared libraries using a protocol specific for small/medium size RNAs. Melanoma and iEV samples appeared well-separated in PCA analysis (Appendix A); mRNA fragments from coding genes were highly represented and equally distributed among the two types of samples (Appendix A); then, we concentrated our first analysis on ncRNAs.

Similar biotypes of ncRNAs were retrieved in both melanoma tissue and in their iEVs (Appendix A), with a larger number of lincRNA transcripts in iEVs than in melanoma. A total of 118 transcripts was differentially represented in melanoma and iEV samples (Figure 3A,B and Appendix A), of which only 17 were overrepresented in iEVs (Figure 3A–D). Among the species enriched in iEVs, lincRNAs and ribozyme RNAs were the most enriched (Figure 3E). Next, we validated some differentially expressed ncRNAs using qPCR. We selected the MRP and P RNAse, as these were the most highly enriched in iEVs, and unlike other ncRNAs, they have been rarely reported as being enriched in EVs. We found that also by qPCR, P and MRP RNAses were enriched in iEVs compared with melanoma (Figure 4A). To check if the enrichment in P and MRP RNAs was specific to melanoma iEVs, we analysed their expression in iEVs derived from a zebrafish brain tumour model induced by the same oncogene [29]. We extracted iEVs from brain tumours using the same protocol used for melanoma tissue and used image stream flowcytometry and NTA to characterize them (Appendix A). Brain tumour iEVs showed the same average size and percent of GFP-fluorescent particles over the total number of iEVs as melanoma iEVs. However, no enrichment for P and MRP was found in the iEVs isolated from brain tumours (Figure 4B), thus suggesting that the enrichment of these ncRNAs in iEVs is specific for melanoma. The human orthologs of these ncRNAs were also enriched in EVs isolated from the culture medium of human melanoma A375 cells compared with their levels in the cultured cells (Figure 4C). To gain insights into whether these iEV-associated ncRNAs are fragmented as compared to P and MRP present in cellular protein complexes, we performed UV crosslinking of RNA–protein complexes before EV extraction, followed by size (30 kDa cut off). The majority of P and MRP RNA in the EVs could be amplified from the high molecular weight fraction (Figure 4D), indicating that they are mostly unfragmented and perhaps complexed with proteins in melanoma EVs. Finally, we tested whether these ncRNAs could be digested by RNAseA or RNAseA and proteinase K treatment. We found that the majority of P and MRP (67 and 57%, respectively) RNAs was digested by RNAseA treatment, a further 12 and 17% required pretreatment with proteinase K before RNAseA treatment, and only 20% of P and 24% of MRP were protected from enzymatic digestion (Figure 4E). This experiment suggested that either most EVs are damaged, thus allowing the penetration of the enzymes, or that most of the P and MRP complexes are readily accessible to the enzymes as they are located on the external surface of the EVs.

### 2.4. Interstitial Melanoma EVs as Mediator of Inflammation through Their RNA Content

Extracellular vesicles released from tumour cells are considered important mediators of intercellular communication. Their ability to activate the innate immune system through danger-associated molecular patterns (DAMPs) is of great interest for melanoma, considering the progresses of tumour immunotherapy in this cancer type. Here we investigated whether iEVs produced by zebrafish melanoma are able to activate the innate immune system by injecting 4 nL of PBS containing 10^9^ iEVs/mL (approximately 4000 iEVs) in the blood stream of 2 days post fertilization (dpf) transgenic fish with fluorescent macrophages (*tg(mpeg:GFP)*^*gl*222^ (Figure 5A)) or neutrophils (*tg(mpx:GFP)*^*i*113^ (Appendix A).

The use of these transgenic lines allowed for the direct evaluation of the responses of innate immune cells to the iEVs. Three days later (at 5dpf), we counted the number of macrophages or neutrophils in the iEV-injected *tg(mpeg:GFP)*^*gl*222^ or *tg(mpx:GFP)*^*i*113^ larvae and compared them with controls, injected with PBS alone. The counts showed that in iEV-injected larvae, there was a significant increase in the number of macrophages (Figure 5B–D), but not neutrophils (Appendix A) compared to PBS-only-injected larvae, and that this increase was accompanied by changes in the morphology of macrophages, which appeared less dendritic than in PBS-only-injected larvae (Figure 5B’,C’). We then evaluated whether injection of iEVs was also able to induce an increase in the expression of inflammatory cytokines, *Il1-β* and *Il-8*, by QPCR analysis of whole larvae.

The expression of *Il8* was significantly increased in iEV-injected larvae, while the increase of the expression of *il1-β* was not significant (Figure 5E). Next, we wanted to see if the ability of iEVs to elicit a macrophage response was due to their ncRNA content, and specifically to P-and/or MRP-RNA. We amplified the ncRNA genes with RT-PCR and subcloned them in a vector suitable for in vitro transcription. In vitro transcribed RNAs encoding for P or MRP were injected in the blood stream of 2 dpf *tg(mpeg:GFP)*^*gl*222^ transgenic larvae (Figure 6A) at a dosage that was approximately 100 times higher than P and MRP contained in 4000 iEVs. As controls, we injected poly-(I:C), a known polynucleotide analog and inducer of sterile inflammation. Three days later (at 5dpf), we counted the number of macrophages in the injected *tg(mpeg:GFP)*^*gl*222^ larvae (Figure 6B–G). Counts showed that P and MRP RNAs injected in the blood stream were able to induce an increase of macrophage number in the tail area similar (P-RNA) or greater (MRP-RNA) to iEVs or poly (I:C). Moreover, all these factors induced a significant increase of the expression of two interferon-responsive genes (IRGs), *ifit10* and *isg15* (Figure 6H), which are often increased in response to poly-(I:C) exposure.

In conclusion, this set of experiments showed that melanoma iEVs and the ribozyme-associated RNAs, P and MRP, highly enriched in iEVs, elicit an inflammatory response when injected in the blood stream, which involves macrophages and the increased expression of IRGs.

## 3. Discussion

Here, we describe a method to extract and characterize interstitial EVs from zebrafish solid tumours using a well-characterized melanoma model [27]. For this purpose, we exploited the NBI isolation method [7] adapted for the purpose in this project. Using this approach, we obtained effective isolation of a heterogenous population of vesicles from zebrafish melanomas and high yields in terms of concentration and quality of iEVs. We also document that purified iEVs are GFP+, which likely results from the integration of the GFP-RAS fusion oncogene within these structures. It must be noted that, so far, most of the attention in characterizing EV function and content has been restricted to circulating exosomes. This is principally due to the fact that these membranous structures are released in all biological fluids, a feature that allows them to be used as potential biomarkers for diagnostic purposes in different pathological conditions, including cancer. However, EVs in biological fluids may have multiple origins, which may dilute or confound their diagnostic or prognostic value, unless they could be compared with EVs directly produced from the tumour cells. Only a few methods to isolate interstitial EVs from whole tissues has been described so far. One method, developed by Hurwitz et al. [25], describes the isolation of small iEVs subpopulations by combining differential ultracentrifugation and small-scale iodixanol density-gradient purification. Teng et al. [26], instead, described a biotin–streptavidin-based detection approach to selectively isolate tumour EVs. This latter method requires the engineering of cancer cells with specific constructs and subsequent administration in immunocompromised animal models. Therefore, these approaches are not ideal for spontaneously-arising tumours, such as our model, or for patient biopsies. Furthermore, both methods cannot exclude potential contamination with EVs deriving from intracellular compartments. In our analysis, we observed the integrity of melanoma cells after single-cell dissociation from the dissected tissue, which prompted us to consider iEVs as secreted particles.

In the cancer field, the major contribution of EVs to tumour progression was initially ascribed to the formation of the pre-metastatic niche, following a pioneering study by Hoshino et al. [20], who demonstrated how circulating tumour-derived exosomes influenced differential metastatic organotropism to distant sites. Here we show that melanoma iEVs and their ncRNA cargo are mediators of sterile inflammation, an event that may have profound effects on local and systemic responses to cancer progression and metastasis. The ability of macrophages to engulf circulating EVs has been documented in several studies, including the beautiful live imaging performed by the Goetz group [4]; however, not many studies have gone on to characterize the effects elicited by tumour EVs on macrophage activities, nor investigated the EV cargos responsible for such effects. Here we were thrilled to find that two ncRNAs, with well-known roles in tRNA and rRNA processing as ribonucleoprotein complexes, RNAse P and RNAse MRP, were enriched in melanoma iEVs. While, approximately 20% of the P and MRP RNAs could be located inside of the iEVs, as they were protected by the RNAse A treatment, the majority of these ncRNA molecules seem to be complexed with protein and possibly localized on the external surface of the iEVs, although further experiments are necessary to confirm these observations.

Why do these ncRNAs, usually located in the nuclear compartment and complexed with several protein subunits, end up in the extracellular vesicles? Through NGS and qPCR analyses, we know that these ncRNAs are present as intact molecules, and although we do not have information about whether the protein subunits of P and MRP ribonucleoprotein complexes (RNC) are also present inside melanoma iEVs, it is known that when the P and MRP RNAs are complexed with their protein subunits, they are promptly transported to the nuclear compartment [30]. Therefore, we hypothesize that these ncRNAs in EVs are possibly complexed with a different set of proteins. Indeed, it would be interesting to investigate whether an uncoupling between the RNA and protein subunits of the P and MRP ribozymes takes place in melanoma cells and is responsible for the accumulation of P and MRP ncRNAs in the EVs. The possible involvement of Pop proteins, which represent 6 of the 10 protein subunits present in P and MRP RNAses [31] in the assembly of the telomerase holoenzyme, through their binding to TERC, the RNA component of telomerase, and their role in telomerase activation, has recently been shown in yeast [32], where pop proteins are necessary for telomerase activation. The hypothesis that in zebrafish melanoma a reactivation of telomerase, that we documented in a previous report [33], involves pop proteins, creating an unbalance between the RNA and protein components of P and MRP RNAses, is an interesting avenue to pursue in future studies. It is intriguing that in a different tumour model, induced by the same oncogene but not associated with a reactivation of telomerase because of the deployment of ALT (Alternative mechanisms of Telomere maintenance, [33]), P and MRP are not found enriched in iEVs.

Finally, we noticed that the inflammation induced by iEVs results in an increase of IRG expression, which suggests that the RNA sensing pathway could be activated in macrophages engulfing melanoma iEVs. Further investigation of the main inflammatory signals conveyed by melanoma iEVs could be important to devise strategies to control the inflammatory responses and their still unknown consequences for tumour progression and metastasis formation. Furthermore, the use of different models to study the effects of RNA molecules enriched in tumour EVs on cells of the adaptive immune system, including adult zebrafish or mouse, may shed further light on the activation and roles of the RNA sensing pathway in tumour immunity [34,35].

In conclusion, we report a versatile approach for the rapid isolation of iEVs starting from zebrafish melanoma tumours, a technique that can be applied to other cancer models and to human cancer biopsies, opening up the field to a new range of diagnostic tools that could be used in parallel with liquid biopsies to validate and reinforce their diagnostic and prognostic value. The zebrafish has proven to be very useful, as the small size of its tumours and the similarities with human pathology resemble the results that can be obtained with a tumour biopsy. Overall, devising a method for the purification of iEVs from whole tissue will be of particular importance also from a clinical perspective, since interstitial EVs could harbour great diagnostic value, particularly in specimens where molecular information are difficult to obtain from histological evaluation.

## 4. Materials and Methods

### 4.1. Zebrafish Maintenance

Adult zebrafish (*Danio rerio*) were housed at the Department of Cellular, Computational and Integrative biology (CIBIO), University of Trento, and maintained in the Model Organism Facility under standard conditions [36]. Zebrafish studies were performed according to European and Italian law, D.Lgs. 26/2014, authorization 75/2017-PR and 148/2018-PR, to M. C. Mione.

In this study we used the following zebrafish transgenic lines: *Et(kita:Gal4TA4, UAS:mCherry)*^*hzm*1^ [26], here called Kita:Gal4, *Et(zic1-4:Gal4TA4, UAS:mCherry)*^*hzm*5^ [29], here called zic:Gal4, and *Tg(UAS:egfp-HRASV12G)*^*io*006^, here called UAS:RAS [27], *tg(mpx:GFP)*^*i*113*Tg*^ [37], and *tg(mpeg1:GFP)*^*gl*22*Tg*^ [38].

### 4.2. Cell Culture and Cell Lines

A375M cells were maintained in DMEM medium (high glucose) supplemented with 10% FBS, 2 mM L-glutamine, and 2 mM of penicillin/streptomycin. All cell culture reagents were obtained from Gibco. Cells were maintained at 37 °C in a humified atmosphere containing 5% CO_2_. Cell lines were tested regularly for mycoplasma contamination by the Celltech CIBIO facility.

### 4.3. Isolation and Purification of EVs from Cell Conditioned Media

EVs were isolated from cell culture media according to the NBI method [7]. Briefly, cells were grown until at least 70% confluence in complete medium. Then, 24 h before performing the isolation, cells were washed twice with PBS 1X, and medium was changed to serum-free medium. Cell culture medium was then collected and centrifuged at 2800× *g* for 10 min at RT to remove potential cellular debris. To the resulting supernatant, homogenised nickel-functionalized agarose beads [7] were added at a concentration of 25 μL of beads per 1 mL of sample. Samples were incubated for 30 min with gentle rotation (9–12 rotations/minute) using a VWR tube rotator to prevent sedimentation of beads and then centrifuged at 600–800× *g* for 2 min to sediment the EV-containing bead pellet. Then 1.2 X elution buffer was freshly prepared by diluting 5X Solution A (PBS 1X + 16 mM EDTA) and 5X Solution B (PBS 1X + 10 mM NaCl + 225 μM citric acid) in 0.22 μm-filtered PBS 1X. Two-bead volumes of elution buffer were added to the EV-containing bead pellet, and the elution was completed by incubating the samples in a thermomixer at 28 °C for 10 min at 600× *g*. After a final centrifugation step at 1800× *g* for 1 min at RT, EVs were collected from the supernatant. EV concentration and size were evaluated using a Nanosight instrument.

### 4.4. Isolation and Purification of iEVs from Tissue

Zebrafish melanomas were dissected and cut into small fragments and then incubated with collagenase from *Clostridium histolyticum* (2 mg/mL) (Sigma-Aldrich, St. Luis, MO, USA) and DNase I (10 U/mL) (ThermoScientific, Waltham, MA, USA) dissolved in 1 mL of RPMI medium (without supplements and phenol red) (ThermoScientific, Waltham, MA, USA ) for 30 min in gentle shaking using a tube rotator (VWR, Milan, Italy). Samples were then filtered with a 40 μm cell strainer, and cellular and tissue debris were removed following two sequential centrifugation steps consisting of 300× *g* for 10 min and then 2000× *g* for 20 min. The resulting supernatant was used as starting material for iEV isolation following the nickel-based isolation (NBI) method as described [7] with a few modifications. Briefly, the supernatant was diluted 1:4 with sterile PBS, and then 100 μL of homogenised nickel-functionalized agarose beads were added. Samples were incubated for 30 min in gentle shaking using a tube rotator (VWR, Milan, Italy) to prevent sedimentation of beads and then centrifuged at 600–800× *g* for 2 min to sediment the iEV-containing bead pellet. Then, 1.2X elution buffer was freshly prepared by diluting 5X Solution A (PBS 1X + 16 mM EDTA) and 5X Solution B (PBS 1X + 10 mM NaCl + 225 μM citric acid) in sterile PBS. Two-bead volumes of elution buffer were added to the iEV-containing bead pellet, and the elution was completed by incubating the samples in a thermomixer at 28 °C for 10 min at 600× *g*. After a final centrifugation step at 1800× *g* for 1 min at RT, iEVs were collected from the supernatant. iEV concentrations and sizes were characterized using a Nanosight instrument (Malvern Pananalytical, UK). To concentrate iEVs for injection, an ultracentrifugation step was included (110.000× *g*, 2 h at 4 °C in a TLA55 fixed-angle rotor, Beckman Coulter Optima MAX-XP Ultracentrifuge, Pasadena, CA, USA). The pellet was resuspended in 20 μL sterile PBS and used immediately or stored at 4 °C for a max of 2 days before injection.

Differential centrifugation to provide an alternative mean of isolating zebrafish melanoma iEVs for TEM analysis was performed as follows: after filtration through a 40 μm cell strainer, two sequential centrifugation steps at 300× *g* for 10 min and 2000× *g* for 20 min were performed to remove cellular and tissue debris, followed by two centrifugation steps at 18,000× *g* for 60 min at 110,000× *g* for 2 h. The same protocol was used for the isolation of iEVs from zebrafish brain tumours.

### 4.5. ImageStream Acquisition and Analysis

Multispectral imaging flow-cytometric acquisition of iEVs and small particle calibrators was performed using an ImageStream^x^ (ISX) *MKII* (Luminex Corporation, Seattle, WA, USA) with a 60x objective, high gain mode, and low flow rate. Particles were labelled with 0.5 μg/mL sterile PBS Cell Mask Deep Red (ThermoFisher Scientific; Waltham, MA, USA, cat. no. C10046) and incubated at RT for 10 min.

The ISX was equipped with 488, 635, and 405 nm lasers. Two channels (Ch01 and Ch09) were set to brightfield (BF). Channels 6 and 12 were set to side-scatter (SSC), and further fluorescence channels were used for fluorescent iEVs detection. The total number of particles was counted in the Ch11 channel, while the percent of GFP-fluorescent particles over the total was obtained in the Ch2 channel.

The advanced fluidic control of ISX, coupled with the presence of continuously running speed beads, enabled cell/particle enumeration using the “objects per mL” feature within the IDEAS^®^ data analysis software. iEV samples were run at concentrations lower than 10^10^ objects/mL. All samples were acquired using INSPIRE^®^ software (Luminex Corporation, Seattle, WA, USA), with a minimum of 10,000 events collected. Data analyses were performed and spectral compensation matrices produced using ISx Data Exploration and Analysis Software (IDEAS^®^, Luminex Corporation, Seattle, WA, USA). Technical controls, in conjunction with a negative control where iEVs were lysed with Triton X-100, were used.

### 4.6. Nanoparticle Tracking Analysis (NTA) of EVs

EV size and concentration were determined by NanoSight NS300 (Malvern Panalytical, Malvern, UK) according to the manufacturer’s instructions. Resuspended differential centrifugation pellets were diluted to 1 mL with PBS. Polystyrene microspheres (0.1 μm; Malvern Panalytical, Malvern, UK) were used to standardize the camera setting in each experiment (camera level 11). At least four videos of 60 s were recorded for each sample dilution and were analysed using NanoSight NS300 software NTA 3.4 Build 3.4.003 (Malvern Panalytical, Malvern, UK).

### 4.7. iEVs Count by TRSP (Tuneable Resistive Pulse Sensing)

Size and concentration of the iEVs were characterized by RRPS using qNANO (Izon science Lyon, France). At least 2 min recording time was used for each sample. NP150 and NP400 nanopores were used to measure small and large iEVs, respectively. All the qNANO data were recorded and analysed by Izon Control Suite v.3 (Izon science, Lyon, France).

### 4.8. Super Resolution and Confocal Microscopy of iEVs

Images for super resolution microscopy were acquired with an ONI Nanoimager (ONI, Oxford, UK) in TIRF mode equipped with an Olympus UPLAPO x100/1.4 OHR objective and a Hamamatsu ORCA Flash 4.0 camera (Hamamatsu Photonics Italia s.r.l., Milan, Italy). 

Confocal microscopy was carried out with a Leica SP8 inverted confocal microscope (Leica Microsystem, Milan, Italy), using a x63 Leica PlanApo objective and a single focal plane.

### 4.9. Transmission Electron Microscopy

To visualize iEVs by electron microscopy, isolated using the NBI method or ultracentrifugation, the iEV pellets were deposited on carbonated grids and fixed in 2% PFA in 0.1 M phosphate buffer, pH 7.4. The grids were then embedded in methyl-cellulose uranyl acetate 0.4%. All samples were examined with a FEI Tecnai Spirit electron microscope (FEI Company, 5350 NE Dawson Creek Drive, Hillsboro, OR, USA), and digital acquisitions were made with a numeric camera (Quemesa; Soft Imaging System, Emsis, Munster, Germany).

### 4.10. Western Blot Analysis

Western blot analysis was carried out using standard methods. Briefly, cells and EV pellets were lysed on ice with lysis buffer (150 mmol/L NaCl, 50 mmol/L TRIS pH 7.4, 0.25% NP40, 1mM EDTA, 0.1% Triton X100, 0.1% SDS, 0.1 mg/mL phenylmethylsulfonyl fluoride, protease cocktail inhibitor (Merck Life Science, Milan, Italy; cat. no.04693159001), 50 mmol/L NaF, and 10 mmol/L Na_3_O_4_V. For Western blots, equal protein concentrations (quantified with the Bicinchoninic Acid (BCA) method, Merck Life Science, Milan, Italy; cat. no. B9643) were resolved via 12% SDS–PAGE and transferred to PVDF membranes (Biorad Laboratories S.r.L. Rome, Italy). Antibodies used were: anti-ALIX (Covalab, Bron, France; cat. no. PAB 0204, 1:1000); anti-TSG101 (Gene Tex, Irvine, CA, USA; cat. no. GTX70255); anti-calnexin (Enzo, Farmingdale, NY, USA; cat. no. ADI-SPA-860-F 1:1000); anti-GFP (Merck Millipore, Milan, Italy; cat. no. 3580, 1:1000); goat anti-rabbit IgG, HRP (Abcam, Cambridge, UK; cat. no. 6721) or goat anti-mouse IgG, HRP (Abcam, Cambridge, UK; cat. no. 6728), 1:5000. ECL Western blotting substrate (GeneTex, Irvine, CA, USA; Trident fento Western HRP substrate, cat. no. GTX14698) was added before detection with a BioRad Chemidoc XRF+ (Biorad Laboratories S.r.L. Rome, Italy). Full blots are shown in Appendix A.

### 4.11. RNA Extraction, Library Construction, and Illumina Sequencing

RNA isolation was performed on cells and EVs using a Qiagen miRNeasy Mini Kit and RNeasy MinElute Cleanup Kit (Qiagen, Hilden, Germany) according to the manufacturer’s instructions. The quality of RNA was checked with a bioanalyzer

Libraries were prepared with a Clontech (Takara, Kusatsu, Japan) SMARTer smRNA-Seq Kit specific for medium/small RNAs. Sequencing was performed using Illumina (San Diego, CA, USA) Hiseq2500, obtaining 8–13 million reads per library.

### 4.12. NGS Data Analysis

Sequencing raw reads (fastq) were quality checked with FASTQC (version 0.11.3) (www.bioinformatics.babraham.ac.uk/projects/fastqc/, accessed on 18 October 2018). All samples passed the quality standards. Reads were aligned to the reference genome Danio rerio assembly GRCz11 using STAR [39] with recommended options and thresholds (version 2.5). HTSeq-count (version 0.9.1) [40] was used to generate raw gene counts. The EdgeR package (version 3.24.3) [41] was used to normalize counts to Trimmed Mean of M-values (TMM) for visualization methods. The differential expression analysis was performed using the DESeq2 package (version 1.22.2) [42], and for significance testing, the Wald test was used. Genes were considered differentially expressed with adjusted *p*-value < 0.05 and a log2 fold change greater than 1 or smaller than −1. Principal component analysis was calculated using the prcomp function. Plots were generated with the packages matplotlib [43] and ggplot2 [44].

Data are available at GEO (acc. no. GSE189352).

### 4.13. Q-PCR Analysis of ncRNA

Total RNA was extracted from iEVs using Trizol (InVitrogen, Waltham, MA, USA) and then isolated using a NORGEN Single Cell RNA Isolation kit (cat. no. 51800) according to manufacturer’s instructions. cDNA was prepared using Superscript IV (Invitrogen, cat. no. 18090010) using a mixture of reverse primers specific for the genes listed below, each at a concentration of 1 μM, according to the manufacturer’s protocol. For RNAse A treatment, 10 ng/uL of RNAse A (Roche, cat no 10103142001) was added after iEV isolation with NBI bead dissociation (for RNA external to iEVs) and incubated for 20 min at 37 °C. For proteinase K (PK) treatment, PK (Promega, cat. no. A505B) was added at a concentration of 1 mg/mL for 10 min at 37 °C and deactivated for 10 min at 65 °C prior to RNAseA. qRT-PCR analysis was performed using SMOBio qPCR Syber Green Mix (TQI-201-PCR Biosystem) using a standard amplification protocol. The primers used are the following: zebrafish *P RNAse* forward 5′ CGAGTGCACAGAAAATCGGG-3′ and reverse 5′-GGGAGAGTAGTCTGAATTGG-3′; Zebrafish *MRP RNAse* forward 5′-GACACTGGGCAGAGATGT-3′ and reverse 5′-CGTATGCTGAGAATGAGCC-3′; zebrafish *rnu 6.1* forward 5′GCTCGCTACGGTGGCACATA-3′ and reverse 5′-CAGCATATGGAGCGCTTCACG-3′; human *RMRPP5* RNAse forward 5′-ACTTGTGCTGAGGACCTGTT-3′ and reverse 5′-GTGTAGTTGGTGCATGTGGG-3′; human *P RNAse* forward 5′-CTGTCACTCCACTCCCATGT-3′ and reverse 5′-ATTGAACTCACTTCGCTGGC-3′; human *RMRP* forward 5′-GGCTACACACTGAGGACTCT-3′ and reverse 5′-TGCGTAACTAGAGGGAGCTG-3′; human *RNU 6.1* forward 5′-CTCGCTTCGGCAGCACA-3 and reverse 5′-GGAACGCTTCACGAATTTGC-3′.

For quantification of absolute amounts of P and MRP RNAs in samples, we constructed an amplification curve based on known amounts of MRP cDNAs using increasing concentrations from 0.001 pg to 1 ng of the plasmid described in Section 4.15 as templates.

Data analysis was performed with Microsoft Excel and GraphPad Prism. In all cases, each qPCR was performed with triplicate samples and repeated with at least three independent samples. Data are expressed as fold changes compared to controls.

### 4.14. Isolation and Quantification of P and MRP in Complexes and as Fragmented RNA in EVs

A375M cells were grown until at least 70% confluence in complete medium. Then, 24 h before performing EV isolation, cells were washed twice with PBS 1X, and medium was changed to serum-free medium. After 24h, UV crosslinking was performed on grown cells and serum-free medium using a UV irradiation chamber (0.150 joule, 20 s, on ice). Fragmented RNA, full length RNA, and RNA complexes were then collected from EVs as follows: EVs were collected as described in the previous paragraph up to the centrifugation step at 800× *g* for 3 min to sediment the EV-containing bead pellet. The supernatant was then removed, and the bead pellet was resuspended in an equal volume of lysis buffer (100mM KCl, 5mM MgCl_2_, 10 mM HEPES pH 7, 0,5% Igepal NP-40, 1 mM DTT, 100U/mL RNase Inhibitors 1x Protease Inhibitor cocktail). Samples were stored at −80 °C for 1 h and then centrifuged 1000× *g* for 5′ (4 °C). Lysate (supernatant) was collected, transferred to Vivaspin20 (30 kDa MWCO, cat no. 289322361, Cytiva, Marlborough, CA, USA), and centrifuged 3500× *g* for 2′ for each 100 mL of lysate. Half of the volume of lysate was passed through the filter (fragmented RNA, less than 100 nt in size), and the other half stayed on the filter (>100 nt RNA, protein–RNA complexes); both phases were collected. The RNA of both phases was then isolated by resuspending lysates in TripleXtractor reagent for RNA isolation (Grisp, cat no. GB23) and using a NORGEN Single Cell RNA Isolation kit (cat. no. 51800), according to the manufacturer’s instructions. cDNA was prepared from equal amounts of total RNA. as described in Section 4.13, followed by Q-PCR analysis for human P and RMRP RNAse and for RNU6.1.

### 4.15. Cloning, In Vitro Transcription, and Zebrafish Injection of P and MRP RNAs

Zebrafish P and MRP RNAse were reverse-transcribed from total RNA extracted from 48 hpf zebrafish larvae using the following primers: for zebrafish P RNAse forward 5′AGCGGAAGGAAGCTCACTG-3′ and reverse 5′-GGGAGAGTA TCTGAATTGG-3′; for zebrafish MRP RNAse forward 5′-GCCATGCCTGAAAACTTG-3′ and reverse 5′-CGTATGCTGAGAATGAGCC-3′ and cloned in PCR 2.1 (Invitrogen, cat. no.46-0641) following the manufacturer’s instructions. Correct clones and orientations were confirmed by sequencing. For in vitro transcription of sense RNA, the Ambion SP6 mMessage machine kit (Ambion, cat. no.AM 1340, Austin, TX, USA) was used.

### 4.16. Injections in 48 hpf Zebrafish and Macrophage/Granulocyte Counts

For injections of iEVs, poly (I:C), and mRNAs, we used volumes of 4 nL of a purified RNA solution containing 0.4 ng of RNA injected in the caudal vein of anaesthetized 48 hpf zebrafish larvae using a glass capillary connected to a FemtoJet microinjector (Eppendorf, Hamburg, Germany), as described in [45].

Other injections were performed with: poly (I:C) (high molecular weight polyinosine-polycytidylic acid, InVivoGen, San Diego, CA, USA, cat. no. tlrl-pic) or GFP mRNA, all at 1 ng/nL in PBS (4 nL).

### 4.17. Statistical Analysis

In all cases, n numbers refer to biological replicates unless otherwise stated. All the graphs and the statistical analyses were generated and calculated using GraphPad Prism software version 5.0. All the graphs represent mean +/− SEM, and data were analysed against controls with unpaired t-tests (no Gaussian distribution, two-tailed, an interval of confidence: 95%), except for different details reported in the figure legends.

## Figures and Tables

**Figure 1 ijms-23-05510-f001:**
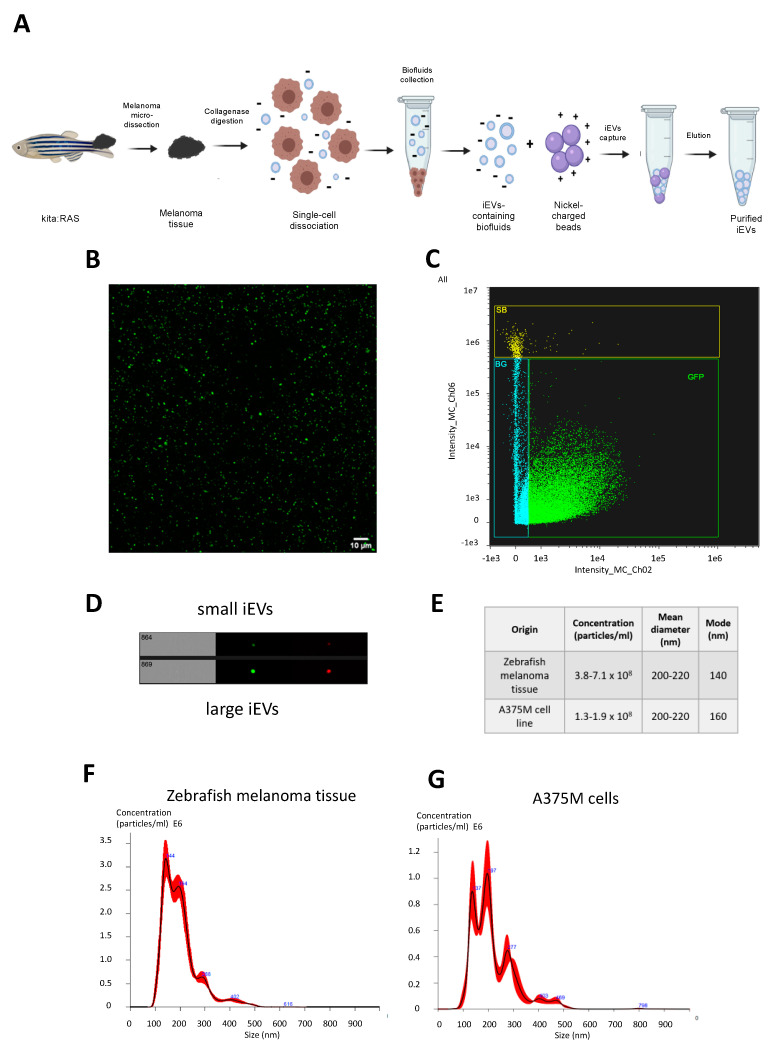
(**A**) Schematic representation of melanoma microdissection, single cells dissociation, and iEVs isolation following the NBI method. (**B**) Purified EVs from kita:RAS zebrafish melanomas (confocal microscopy, 63x magnification). (**C**) Representative sorting of iEVs isolated from kita:RAS zebrafish melanomas (ImageStream flow cytometer), and (**D**) single images of particles at the same instrument. (**E**) Table reporting concentration, mean, and mode diameter values retrieved from NTA performed on three independent biological samples from either kita:RAS zebrafish melanomas or the A375M human melanoma cell line. (**F**) Representative nanoparticle tracking analysis (NTA) profiles displaying concentration (particles/mL) vs. size distribution (nm) of iEVs isolated from kita:RAS zebrafish melanoma, and (**G**) EVs isolated from the cell culture supernatant of the A375M human melanoma cell line. The black curve indicates the mean of three measurements, with SD in red.

**Figure 2 ijms-23-05510-f002:**
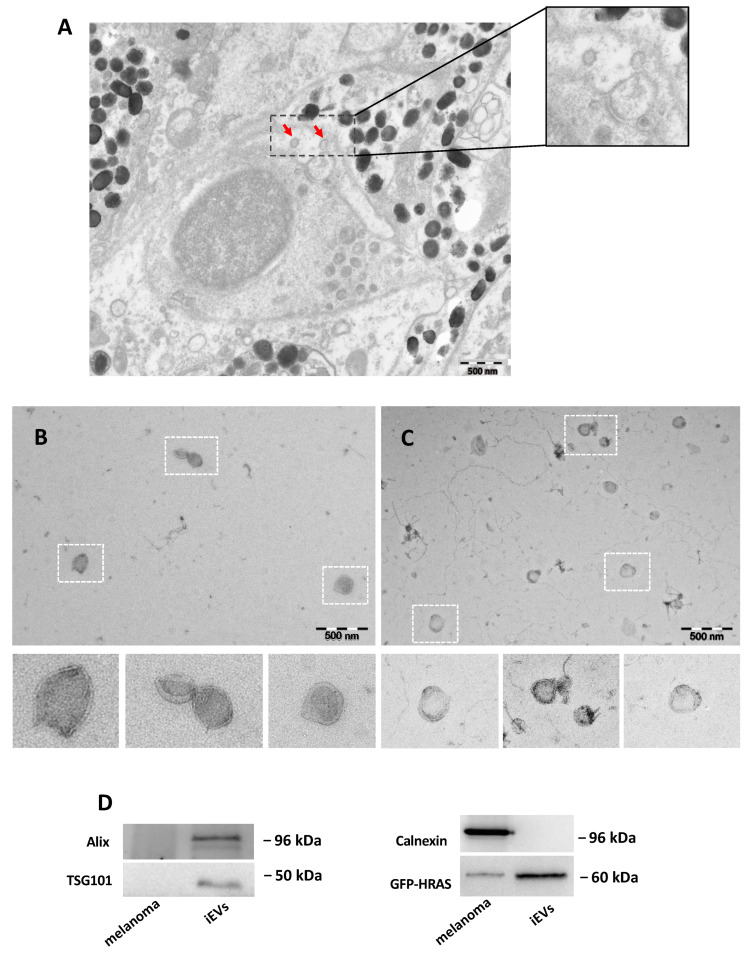
Characterization of zebrafish melanoma iEVs: (**A**) representative EM of zebrafish melanoma, with EVs in the tissue (red arrow); (**B**) representative TEM image of zebrafish melanoma iEVs purified following the NBI method. Scale bar: 500 nm. (**C**) Representative TEM image of zebrafish melanoma iEVs purified following the differential ultracentrifugation method. Scale bar: 500 nm. (**D**) Western blot analysis of whole cell lysates and lysates of isolated iEVs for exosomal-related markers or cell markers, as indicated. Loading: 1 μg total protein.

**Figure 3 ijms-23-05510-f003:**
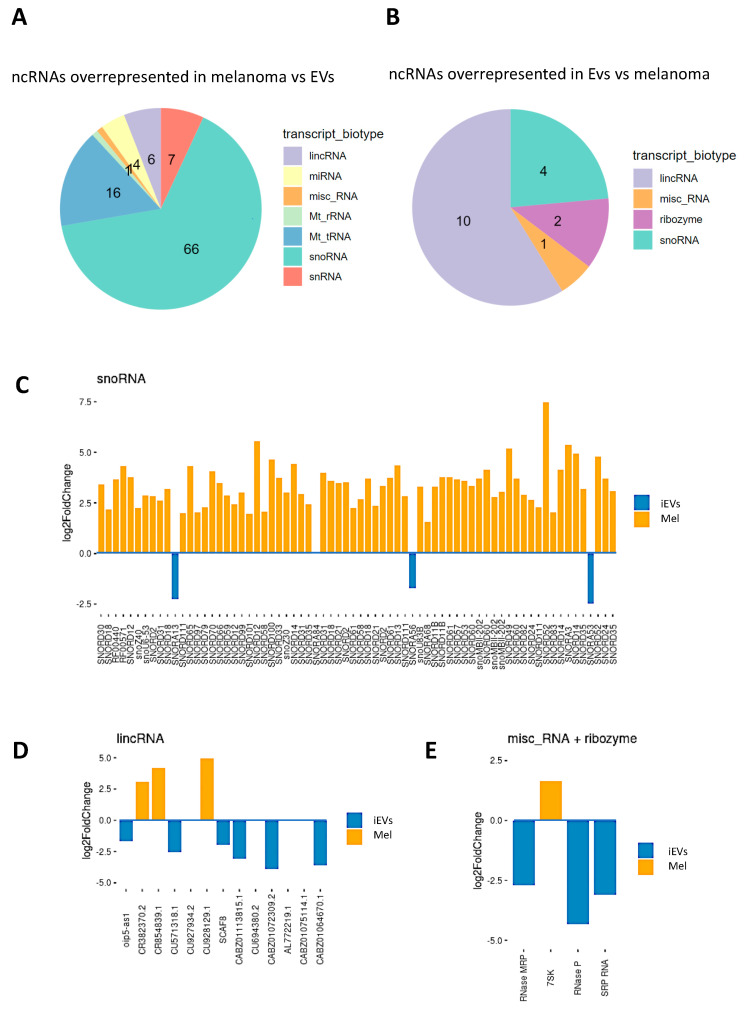
Analysis of ncRNA in iEVs and melanoma. (**A**,**B**) Pie charts representing the different ncRNA species found enriched in melanoma (**A**) and in their iEVs (**B**). ncRNAs were considered differentially expressed with adjusted *p*-value < 0.05 and a log2 fold change greater than 1 or smaller than −1. (**C**–**E**) Representation of the three main categories of differentially enriched species of ncRNAs (orange bars: enriched in melanoma, blue bars, enriched in iEVs) and their identity. Misc_RNA: miscellaneous RNA.

**Figure 4 ijms-23-05510-f004:**
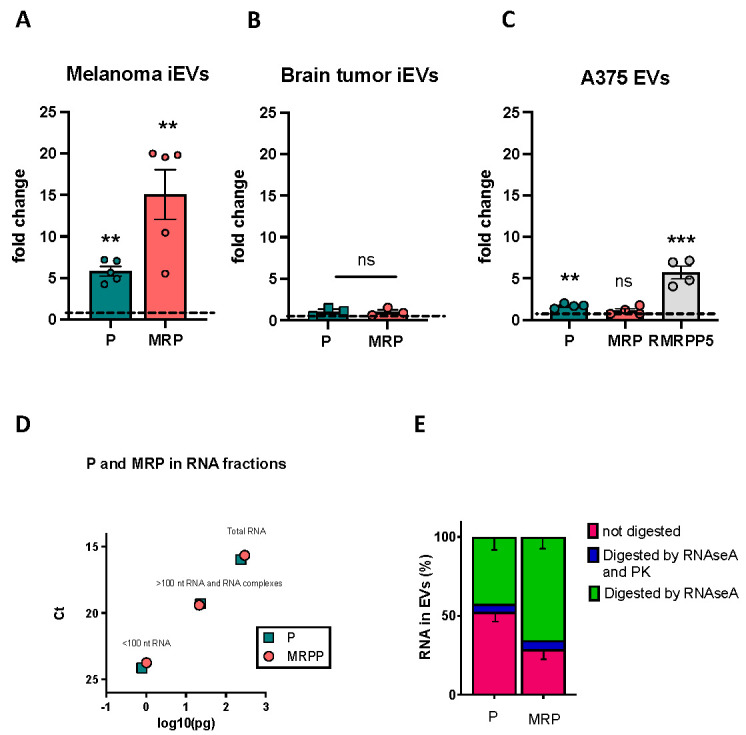
QPCR validation of P and MRP RNAs enrichment in iEVs. (**A**–**C**) Expression of P and MRP (and RMRPP5 in human cells/EVs) RNA in melanoma iEVs compared to the expression in melanoma (**A**), in brain tumour iEVs compared to the expression in tumour cells (**B**), and in A375 EVs versus A375 cells (**C**). In all graphs, expression in the cells of origin of the iEVs/EVs is set at 1, and data are normalized to the levels of rnu6.1/RNU6. n > 3 as represented; ** *p* < 0.01; *** *p* < 0.001. (**D**) Graphic representation of the amount of P and MRP found in the lower fraction (fragmented RNA), higher fraction (RNA and RNA–protein complexes), and total RNA of A375 EVs, following UV crosslinking. Values are expressed in pg and are calculated from the number of ct of a reference scale obtained with serial dilutions of P and MRP cDNA templates (see Section 4.16). (**E**) Percentages of P and MRP RNA lost in RNAseA digested and RNAseA + proteinase K digested in A375 EVs, as indicated. Data are from 3 biological replicates +/− SD. ns: not significant.

**Figure 5 ijms-23-05510-f005:**
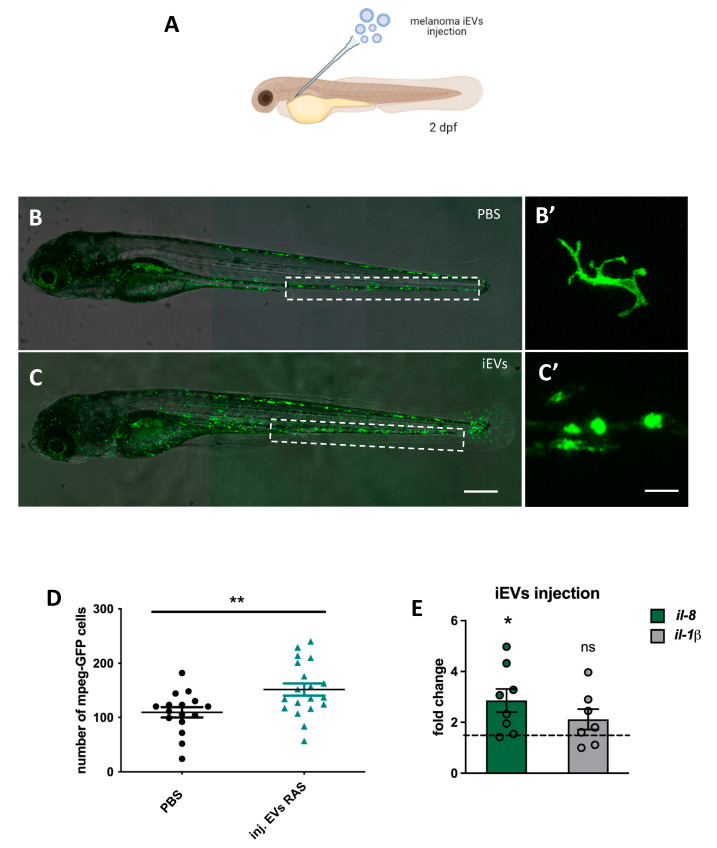
Intravascular injection of iEVs induce an inflammatory response. (**A**) Schematic representation of the experimental procedure for intravascular injection of melanoma iEVs (drawing realized with BioRender). (**B**,**C**) Fluorescence stereomicroscope images of 5 dpf *tg(mpeg:GFP)*^*gl*222^ larvae injected intravascularly with PBS or with melanoma iEVs at 2dpf, as indicated. The caudal hematopoietic tissue (CHT), where counts were performed, is boxed. (**B’**,**C’**) High magnification images of macrophages in the injected *tg(mpeg:GFP)*^*gl*222^ showing differences in shape in the two conditions. (**D**) Counts of macrophages in the CHT of injected larvae. (**E**) QPCR analysis of the expression of two inflammatory cytokines in iEV-injected larvae compared to PBS-injected (horizontal dashed line). * *p* < 0.05; ** *p* < 0.01. Scale bars: 100 μm. ns: not significant.

**Figure 6 ijms-23-05510-f006:**
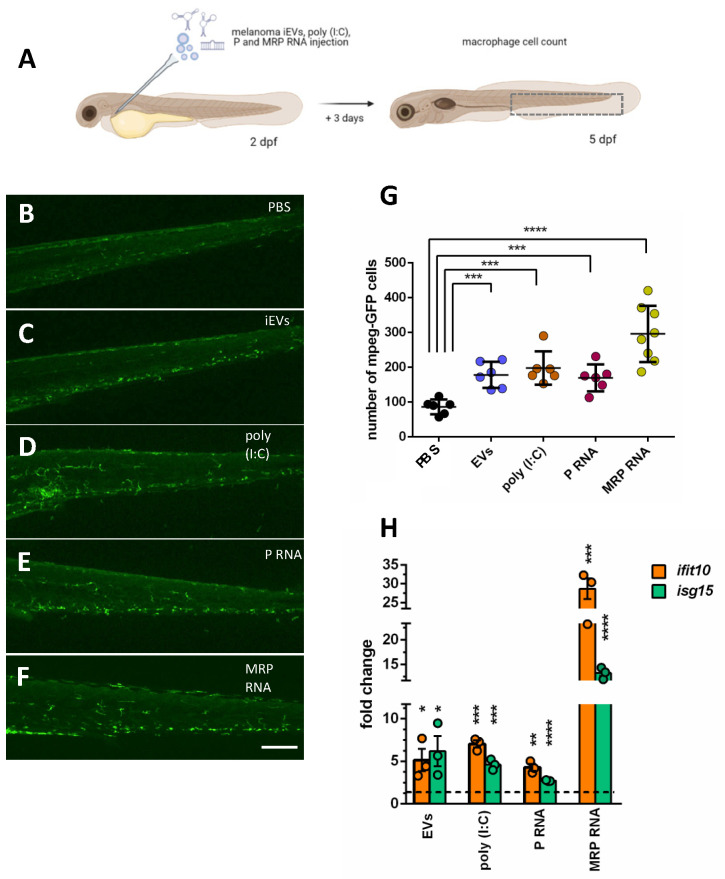
Injection of MRP or P RNA in larvae increases mpeg+ cells and IRG expression, similar to poly-I:C and iEVs. (**A**) Schematic representation of the experimental procedure for intravascular injection of melanoma iEVs, poly (I:C) and RNA, and inflammatory cell counts (drawing realized with BioRender). (**B**–**F**). Representative images of the CHT regions of 5 dpf *tg(mpeg:GFP)*^*gl*222^ larvae injected with the indicated molecules. (**G**)). Counts of macrophages in the CHT of injected larvae. (**H**) QPCR analysis of the expression of two interferon-responsive genes, *ifit10* and *isg15*, in larvae injected with the indicated molecules, compared to PBS-injected (horizontal dashed line). * *p* < 0.05; ** *p* < 0.01; *** *p* < 0.005; **** *p* < 0.001. Scale bars: 100 μm.

## Data Availability

NGS data supporting the reported results can be found at GEO (acc. no. GSE189352).

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
