# Peer review of "Zebrafish Melanoma-Derived Interstitial EVs Are Carriers of ncRNAs That Induce Inflammation"

_ijms, 2022, doi:10.3390/ijms23105510_

Round 1

Reviewer 1 Report

The article by Biagini et al. “Zebrafish melanoma-derived interstitial EVs are carriers of 2 ncRNAs that induce inflammation” describes an optimized method for isolation and characterization of zebrafish melanoma interstitial extracellular vesicles. The work presents valuable information and I find the study well designed and executed. I do not have any major concerns regarding the publication of this work with exception to few small comments mostly concerning the figures that the authors might wish to consider.

  1. There is a “?” sign, which appears throughout Figure 1 (and in some other figures), probably there due to formatting but it makes it difficult to read the information in some places.
  2. The drawings in Figure 1A should be described below each one, because it is not so obvious for every reader what is depicted there.
  3. Axis labels in Figure 1C, E and F are too small and cannot be read.
  4. I would completely remove panel D from Figure 1 or place it in the supplementary information.
  5. The scale bars in Figure 2B and C are not readable; the same is true for the molecular weights in panel D.
  6. The legends next to the pie charts in Figure 3A and B are too small and hard to read.
  7. In Figure 4 “fold changes” should be “fold change”.
  8. In Figure 6 the “G” is hidden.
  9. The authors use some abbreviations that must be clarified or written fully: misc_RNA, cond, ns, dpf, high mag.

Author Response

The article by Biagini et al. “Zebrafish melanoma-derived interstitial EVs are carriers of 2 ncRNAs that induce inflammation” describes an optimized method for isolation and characterization of zebrafish melanoma interstitial extracellular vesicles. The work presents valuable information and I find the study well designed and executed. I do not have any major concerns regarding the publication of this work with exception to few small comments mostly concerning the figures that the authors might wish to consider.

 We thank the reviewer for her/his positive assessment of our ms. Please find below our point-to point response.

  1. There is a “?” sign, which appears throughout Figure 1 (and in some other figures), probably there due to formatting but it makes it difficult to read the information in some places.

Unfortunately, I cannot see this problem in any version downloaded from the web site (ijms-1701822.docx or ijms-1701822.pdf), it may be related to the version that the journal sent to the reviewer. Can the editor please check and let me know how to proceed?

  1. The drawings in Figure 1A should be described below each one, because it is not so obvious for every reader what is depicted there.

We added a short description of the steps depicted in figure 1A under each icon (FEDERICA to do)

  1. Axis labels in Figure 1C, E and F are too small and cannot be read.

We increased the size of the labels in Fig. 1C, E and F

  1. I would completely remove panel D from Figure 1 or place it in the supplementary information.

We have moved fig. 1D to Supplementary fig. S1E

  1. The scale bars in Figure 2B and C are not readable; the same is true for the molecular weights in panel D.

We have increased the size of the labels in the scale bars of fig. 2B and C and for the molecular weights in panel D.

  1. The legends next to the pie charts in Figure 3A and B are too small and hard to read.

We have increased the size of the legends

  1. In Figure 4 “fold changes” should be “fold change”.

We have changed “fold changes” to “fold change”

  1. In Figure 6 the “G” is hidden.

We corrected the mistake

  1. The authors use some abbreviations that must be clarified or written fully: misc_RNA, cond, ns, dpf, high mag.

We have added the full names next to the abbreviation at the first appearance.

Reviewer 2 Report

The aim of the study described in this manuscript was to optimize a method for the isolation of zebrafish melanoma-derived interstitial extracellular vesicles (iEVs) and iEVs subsequent characteristics in terms of concentration, size, EV-related markers, ncRNA content and ability of inducing an inflammatory response.

My comments are as follows:

  1. It is generally accepted that the size of ectosomes typically range from 100 nm up to 1 μm in diameter. I wonder why the authors give a different diameter range for this sub-population of EVs, i.e. 50-1000 nm (lane 56).
  2. The articles listed as References # 8 and # 9 were published in 2009 and 2011 respectively. I propose to place newer reviews instead of them, which will better present the current state of knowledge regarding EVs.
  3. In my opinion the title of 4.3 (lane 390) should be changed. The description shows that EVs were isolated not from the cell supernatant but from A375M conditioned media.
  4. By analyzing the EVs and iEVs isolation protocols from A375M conditioned media and kit: RAS fishes, respectively, I noticed that after competitive EVs / iEVs-beads dissociation, EVs and iEVs present in the respective solutions were centrifuged. My guess is that the given centrifugation conditions for EVs (a final centrifugation step at 1800 rpm for 1 minute at RT (lane 404)) and for iEVs (a final centrifugation step at 600-800 x g for 30 seconds at RT (lane 424)) are relate to pelleting agarose beads. So since EVs and iEVs were, I quote "collected from supernatants", the isolation protocol description lacks what this collection was about. Were supernantants containing EVs or iEVs additionally centrifuged before EVs and iEVs concentration and size were characterized?
  5. In 4.3. and 4.4., please convert all rpm to g.
  6. Why are there two values each time in the Mean, SD as well as CV columns in Fig. 1D? There is no information as to what they refer to. Is it about small and large iEVs?
  7. If the size of exosomes typically range from 40 nm up to 150 nm in diameter, then based on the results shown in Fig. 1EG, I cannot agree with the statement that large quantity of iEVs is mostly in the size range of exosomes (lanes143-144).
  8. For reasons mentioned in point 8, I think that in 4.9 lane 475 instead of "exosome pellets" should be "iEVs pellets".
  9. In 4.10, please add by what method the protein concentration was determined.
  10. Perhaps this issue only affects my pdf file, but I got a lot of question marks in Fig. 1 and Fig. 2.
  11. Font size seems too small to me in Fig. 1 and Fig. 3.
  12. In Fig. 3B, should be “EVs” instead of “Evs”. Moreover, since the analysis presented in Fig. 3 concerns iEVs, consequently all “EVs” should be converted to “iEVs” to avoid confusion. In this manuscript, abbreviation “EVs” stands for extracellular vesicles isolated from A373M conditioned media.
  13. Lanes: 190-218 and 268, please replace all "fig" with "Fig.".
  14. Lane 211, please replace “KDa” with “kDa”.
  15. Lane 214, please replace “proteinase treatment” with “proteinase K treatment”.
  16. In Fig. 4ABC, if all plots had the same scale (Y axis up to 25), the fold change would be more visible for each type of EVs (melanoma iEVs, brain iEVs, A375 EVs).
  17. Fig. 6 legend, please start the description with a capital letter.

Author Response

Comments and Suggestions for Authors

The aim of the study described in this manuscript was to optimize a method for the isolation of zebrafish melanoma-derived interstitial extracellular vesicles (iEVs) and iEVs subsequent characteristics in terms of concentration, size, EV-related markers, ncRNA content and ability of inducing an inflammatory response.

We thank the reviewer for her/his assessment of our ms. Please find below our point-to point response.

My comments are as follows:

  1. It is generally accepted that the size of ectosomes typically range from 100 nm up to 1 μm in diameter. I wonder why the authors give a different diameter range for this sub-population of EVs, i.e. 50-1000 nm (lane 56).

We have corrected the range to 100nm - 1 micron

  1. The articles listed as References # 8 and # 9 were published in 2009 and 2011 respectively. I propose to place newer reviews instead of them, which will better present the current state of knowledge regarding EVs

We have added new, more recent references.

  1. In my opinion the title of 4.3 (lane 390) should be changed. The description shows that EVs were isolated not from the cell supernatant but from A375M conditioned media.

We have changed the title of section 4.3 to “4.3. Isolation and Purification of EVs from cell conditioned media”

  1. By analyzing the EVs and iEVs isolation protocols from A375M conditioned media and kit: RAS fishes, respectively, I noticed that after competitive EVs / iEVs-beads dissociation, EVs and iEVs present in the respective solutions were centrifuged. My guess is that the given centrifugation conditions for EVs (a final centrifugation step at 1800 rpm for 1 minute at RT (lane 404)) and for iEVs (a final centrifugation step at 600-800 x g for 30 seconds at RT (lane 424)) are relate to pelleting agarose beads. So since EVs and iEVs were, I quote "collected from supernatants", the isolation protocol description lacks what this collection was about. Were supernantants containing EVs or iEVs additionally centrifuged before EVs and iEVs concentration and size were characterized?

After the centrifugation of the nickel functionalized agarose beads with the EVs or iEVs attached, the pellet containing the beads-iEVs/EVs complexes were subjected to elution as described (lines 401-406 and lines 421-426) : “1.2X Elution Buffer was freshly prepared by diluting 5X Solution A (PBS 1X + 16 mM EDTA) and 5X Solution B (PBS 1X + 10 mM NaCl + 225 μM citric acid) in sterile PBS. 2-bead volumes of Elution Buffer were added to the iEVs-containing bead pellet and the elution was completed by incubating the samples in a thermomixer at 28 °C for 10 minutes at 600 rpm. After a final centrifugation step at 1800 rpm for 1 minute at RT, EVs were collected from the supernatant”.  Therefore, yes after elution there is a further step of 1800 rpm for 1 m to collect the EVs/iEVs from supernatant before EVs and iEVs concentration and size were characterized.

We have corrected a few other mistakes in these paragraphs (all in track change)

  1. In 4.3. and 4.4., please convert all rpm to g.

We have converted rpm to g.

  1. Why are there two values each time in the Mean, SD as well as CV columns in Fig. 1D? There is no information as to what they refer to. Is it about small and large iEVs?

The two values refer to the two channels used for  gating the information, Ch02 and Ch06. We have now explained it in the legend to:

 Fig. 1D now Fig S1E

Table reporting the intensity MC (Masks Combined standard setting) of channel 2 (GFP) and 6 (SSC) for the indicated populations. Mean, standard deviation (Std. Dev.) and coefficient variable (CV) are included: the two values separated by the comma refer to the considered channels.

MC = This MC mask setting applied by the IDEAS analysis software by default represents a Boolean OR logic of all 12 channel masks(M01-M12) according to the default masking algorithm. 

  1. If the size of exosomes typically range from 40 nm up to 150 nm in diameter, then based on the results shown in Fig. 1EG, I cannot agree with the statement that large quantity of iEVs is mostly in the size range of exosomes (lanes143-144).

The mode of the iEVs is 140 nm, which means that the majority of the iEVs are about 140 nm in diameter. (Mode: The most repeated number in a given set of observations or the Number or Value which have the highest frequency in a given series of numbers).

  1. For reasons mentioned in point 8, I think that in 4.9 lane 475 instead of "exosome pellets" should be "iEVs pellets".

We changed “exosome pellet” to “iEV pellet”

  1. In 4.10, please add by what method the protein concentration was determined.

We added the method by which protein concentration was determined (BCA method)

  1. Perhaps this issue only affects my pdf file, but I got a lot of question marks in Fig. 1 and Fig. 2.

Unfortunately, I cannot see this problem in any version downloaded from the web site (ijms-1701822.docx or ijms-1701822.pdf), it may be related to the version that the journal sent to the reviewer. Can the editor please check and let me know how to proceed?

  1. Font size seems too small to me in Fig. 1 and Fig. 3.

We increased the font size in fig. 1 and 3

  1. In Fig. 3B, should be “EVs” instead of “Evs”. Moreover, since the analysis presented in Fig. 3 concerns iEVs, consequently all “EVs” should be converted to “iEVs” to avoid confusion. In this manuscript, abbreviation “EVs” stands for extracellular vesicles isolated from A373M conditioned media.

We have changed all the EVs from melanoma into iEVs

  1. Lanes: 190-218 and 268, please replace all "fig" with "Fig.".

Thank you, done

  1. Lane 211, please replace “KDa” with “kDa”.

Changed.

  1. Lane 214, please replace “proteinase treatment” with “proteinase K treatment”.

Replaced

  1. In Fig. 4ABC, if all plots had the same scale (Y axis up to 25), the fold change would be more visible for each type of EVs (melanoma iEVs, brain iEVs, A375 EVs).

We have now represented all the plots in Fig. 4 A, B and C with the same Y axis

Fig. 6 legend, please start the description with a capital letter.

Done

Reviewer 3 Report

In their paper entitled “Zebrafish melanoma-derived interstitial EVs are carriers of 2 ncRNAs that induce inflammation”, Biagini et al report isolation and characterization of  zebrafish (ZF) melanoma-derived interstitial extracellular vesicles (iEVs). ZF iEVs contained fluorescently tagged oncogene and were enriched in several ncRNAs, most significantly P and MRP RNAses; when injected into ZF larvae, iEVs increased macrophage number and expression of 2 interferon responsive genes similarly to (albeit not as strongly as) purified P and MRP RNAs. Although the exact biological effect of ZF iEVs remains unclear, the results are interesting in that they map out a pathway into studies of ZF immune responses, building upon studies of melanoma cell differentiation, migration and invasion in this model.

Major points

Is it plausible that iEV purification in different models is affected by the enzymatic treatment (i.e. was trypsin concentration, treatment duration and any associated processing matched exactly for ZF melanoma tissue, ZF brain tumor tissue and human melanoma cell line treatment - as any of these may have affected iEC formation)?

Could the authors provide some quantitation of P and MRP RNAs contained in 4000 iEVs (line 227) to connect to Fig 6G-H – it this the dose effect?

The authors chose to concentrate on RNAses. What about the 6 DE x lncRNAs from Fig 3D, anything on their putative biological role? Potential links to other model systems (mouse, human)? Current description (lines 189-193) is too brief and should be expanded.

The results from Fig 3A-E should be provided separately in a Table form (e.g.  Supplementary Excel file)

A consideration of non-ZF models where the adaptive immunity could also be studied (e.g. mouse), should be included in the Discussion section.

Minor

  1. Please ensure correct formatting of Fig 1 (all panels), 2D (multiple [?] marks) and Fig 6G
  2. Any abbreviation in the Abstract should be explained

Author Response

Comments and Suggestions for Authors

In their paper entitled “Zebrafish melanoma-derived interstitial EVs are carriers of 2 ncRNAs that induce inflammation”, Biagini et al report isolation and characterization of  zebrafish (ZF) melanoma-derived interstitial extracellular vesicles (iEVs). ZF iEVs contained fluorescently tagged oncogene and were enriched in several ncRNAs, most significantly P and MRP RNAses; when injected into ZF larvae, iEVs increased macrophage number and expression of 2 interferon responsive genes similarly to (albeit not as strongly as) purified P and MRP RNAs. Although the exact biological effect of ZF iEVs remains unclear, the results are interesting in that they map out a pathway into studies of ZF immune responses, building upon studies of melanoma cell differentiation, migration and invasion in this model.

Major points

Is it plausible that iEV purification in different models is affected by the enzymatic treatment (i.e. was trypsin concentration, treatment duration and any associated processing matched exactly for ZF melanoma tissue, ZF brain tumor tissue and human melanoma cell line treatment - as any of these may have affected iEC formation)?

We used a mix of collagenase (2 mg/ml) and DNase I (10 U/ml) dissolved in 1 ml of RPMI medium for both melanoma and brain tumors with the same incubation time and no enzymatic treatment at all for the A375 condition media. Collagenase should digest mostly the extracellular matrix and we suppose that it may work in a similar way in both tumor types. We checked the tumor cells after this treatment and we found that they were not damaged, as shown in supplementary Fig. 1A, so we are confident that the treatment did not affect the two tissues in a different way, at least for iEVs formation.

Could the authors provide some quantitation of P and MRP RNAs contained in 4000 iEVs (line 227) to connect to Fig 6G-H – it this the dose effect?

Yes, we have provided this information in the text (line 289), the amount of P and MRP RNA injected in larvae was approximately 100 times higher than those contained in 4000 iEVs.

The authors chose to concentrate on RNAses. What about the 6 DE x lncRNAs from Fig 3D, anything on their putative biological role?

We decided to work of these RNAses as their presence in EVs had not been extensively reported before, while at least 1 of the lncRNAs that we found enriched in iEVs have been previously reported and their biological functions investigated (i.e. oip5-AS1, PMID: 34653794 DOI: 10.1016/j.molimm.2021.10.002 ). It is possible that also those lncRNAs may induce inflammation, and we will investigate this activity in future work.

Potential links to other model systems (mouse, human)? Current description (lines 189-193) is too brief and should be expanded.

We have added some consideration about other model systems in the discussion (lines 386-389). For the description in lines 189-193, we have added the cut-off used for the analysis of DE genes in the legend to figure 3.

The results from Fig 3A-E should be provided separately in a Table form (e.g.  Supplementary Excel file)

We had provided those results as Supplementary Table 1 and 2, not sure why the editor did not forward them to the reviewers. We will re-submit them

A consideration of non-ZF models where the adaptive immunity could also be studied (e.g. mouse), should be included in the Discussion section.

We have added some consideration and an additional reference about other model systems in the discussion (lines 386-389).

Minor

  1. Please ensure correct formatting of Fig 1 (all panels), 2D (multiple [?] marks) and Fig 6G

Unfortunately, I cannot see this problem in any version downloaded from the web site (ijms-1701822.docx or ijms-1701822.pdf), it may be related to the version that the journal sent to the reviewer. Can the editor please check and let me know how to proceed?

  1. Any abbreviation in the Abstract should be explained

Done